# Verification and reproducible curation of the BioModels repository

Lucian P. Smith[1]*, Rahuman S. Malik-Sheriff[2], Tung V. N. Nguyen[2], Henning Hermjakob[2], Jonathan Karr[3], Bilal Shaikh[3], Logan Drescher[4], Ion I. Moraru[4], James C. Schaff[4], Eran Agmon[4], Alexander A. Patrie[4], Michael L. Blinov[4], Joseph L. Hellerstein[5], Elebeoba E. May[6], David P. Nickerson[7], John H. Gennari[8], Herbert M. Sauro[1]

1 Department of Bioengineering, University of Washington, Seattle, Washington, United States of America, 2 European Molecular Biology Laboratory, European Bioinformatics Institute (EMBL-EBI), Cambridge, United Kingdom, 3 Icahn School of Medicine at Mount Sinai, New York, New York, United States of America, 4 University of Connecticut School of Medicine, Farmington, Connecticut, United States of America, 5 eScience Institute, University of Washington, Seattle, Washington, United States of America, 6 Department of Medical Microbiology and Wisconsin Institute of Discovery, University of Wisconsin-Madison, Madison, Wisconsin, United States of America, 7 Auckland Bioengineering Institute, University of Auckland, Auckland, New Zealand, 8 Department of Biomedical Informatics and Medical Education, University of Washington, Seattle, Washington, United States of America

* lpsmith@uw.edu

## Abstract

The BioModels Repository contains over 1000 manually curated mechanistic models from published literature, most often encoded in the Systems Biology Markup Language (SBML). This community-based standard formally specifies each model, but does not describe the computational experimental conditions to run a simulation and collect data. Therefore, it can be challenging to reproduce any figure or result from a publication with an SBML model alone. The Simulation Experiment Description Markup Language (SED-ML) provides a solution: a standard way to specify exactly how to run an experiment corresponding to a specific figure or result. BioModels was established years before SED-ML, and both systems evolved over time, both in content and acceptance. Hence, only about half of the entries in BioModels contained SED-ML files, and these files reflected the version of SED-ML that was available at the time. Additionally, almost all of these SED-ML files had at least one minor mistake that made them impossible to run. To make these models and their results more reproducible, we report here on our work updating, correcting and generating new SED-ML files for 1055 curated mechanistic models in BioModels. In addition, because SED-ML is implementation-independent, it can be used for *verification*, demonstrating that results hold across multiple simulation engines. We tested, corrected, and improved over 450 existing SED-ML files in the BioModels database, and created basic files for the rest of the entries. Then, we used a wrapper architecture for interpreting SED-ML, and report verification results across five different ODE-based biosimulation engines, after further improving the models, the wrappers, and

**Data availability statement:** The updated models and files referenced in this work are publicly available at https://biomodels.org/ and code supporting this study is available at https://github.com/sys-bio/temp-biomodels and https://github.com/biosimulations/biosimulations-runutils. In addition, the simulator wrappers are available on at https://github.com/orgs/biosimulators/repositories, with the Docker images of those simulators available at https://biosimulators.org/simulators.

**Funding:** This work was supported by the Center for Reproducible Biomedical Modeling, funded by the NIH grant P41EB023912 to HS and IIM. Additional support came from NIH grant R24GM137787 to IIM, EMBL Core Funding to HH, and the Innovative Medicines Initiative 2 Joint Undertaking under grant agreement number 116030 (TransQST) to HH. This Joint Undertaking receives support from the European Union's Horizon 2020 Research and Innovation Programme and European Federation of Pharmaceutical Industries and Associations (EFPIA). The funders had no role in study design, data collection and analysis, decision to publish, or preparation of the manuscript.

**Competing interests:** The authors have declared that no competing interests exist.

the engines themselves. Our work with SED-ML and the BioModels collection aims to improve the utility of these models by making them more reproducible and credible. Improved reproducibility means these models are now even more fit for re-use, such as in new investigations and as components of multiscale models.

## Author summary

Reproducing computationally-derived scientific results seems like it should be straightforward, but is often elusive. Code is lost, file formats change, and knowledge of what was done is only partially recorded and/or forgotten. Model repositories such as BioModels address this failing in the Systems Biology domain by encoding models in a standard format that can reproduce a figure from the paper from which it was drawn. Here, we delved into the BioModels repository to create and correct the instructions on what to do with every curated model, and then tested those instructions on a variety of simulation platforms, allowing us to find and correct issues with the platforms and the simulators themselves. Not only did this improve the BioModels repository, but also improved the infrastructure necessary to run these verification comparisons in the future, and improved the fitness of the models for re-use by other researchers.

## Introduction

The reproducibility of scientific research is one of the cornerstones of the scientific method. It may therefore be surprising to learn that the reproducibility of scientific work has been of some recent concern [1]. Numerous articles [2] and reports [3] have been published that discuss reproducibility challenges and possible remedies. Still more surprising, reproducibility is frequently an issue in computational studies, even for deterministic calculations. As an example, a recent survey by the BioModels team [4] confirmed previous anecdotal evidence that a large proportion (over 50%) of published computational models of physiological processes were essentially irreproducible based on the information provided in the manuscript.

Here, we focus on the reproducibility and verification of computational models published by the systems biology and physiology communities. These are simulation studies based on a proposed mechanistic model of some biological process. For the purpose of this paper, we consider *reproducibility* to mean that the computational procedures of the paper can be verified by reproducing the execution of the computational model. We use the definition of *verification* from the Los Alamos National Laboratory white paper [5], which defines verification as '...the process of determining that a model implementation accurately represents the developer's conceptual description of the model and the solution to the model.' Practically, for these models, this is accomplished if separate simulation engines produce the same results when running the same computational experiment. Both are key to model fitness for re-use and expansion in new contexts, as all science strives to be. We define *repeatability*

to mean that the computational experiment may be repeated using the same software and computational environment. Although important, repeatability is a much weaker signal of scientific robustness, as unknown errors within the single software environment may be responsible for the reported conclusions.

In theory, the results from computational models should be easily reproducible, as the mathematics involved is well understood, and the calculations should be the same regardless of the particular software or operating system used. However, modeling has been no more immune to reproducibility failures than any other branch of science [1], and attempts to reproduce even well-known models in the field have encountered a wide variety of problems [6].

In this paper, we will examine the reproducibility and verification of the large corpus of computational models held at the BioModels repository [7]. Our studies are greatly facilitated by the Systems Biology Markup Language (SBML) standard that allows for separating the description of a model from its execution. This separation enables comparison of results for the same model on different simulators. A second standard, the Simulation Experiment Description Markup Language (SED-ML), provides an implement independent description of procedures in computational experiments.

## Modeling standards

**SBML**: In the last 15 years, the reuse and long-term storage of computational models generated by the biomedical research community has been greatly improved by the use of community modeling standards [4,8]. One of the more important such standards is the Systems Biology Markup Language (SBML) [9]. SBML is a standardized, machine-readable format for representing computational models of biological processes. The emergence of SBML stimulated the development of repositories for storing models. One of the most well-known of these is BioModels [7] but others exist, such as BiGG [10], KBase [11], and Model SEED [12]. The models stored at BioModels are mostly kinetic models that have been obtained from the literature, and the collection includes metabolic, protein signaling, and gene regulatory models. BioModels has grown to include almost 1100 curated models (as of May 2025) collected from peer reviewed articles. These models have been *manually* curated by staff at the European Bioinformatics Institute (EMBL-EBI) as well as by collaborators at the University of Hertfordshire, the California Institute of Technology, and other institutions to ensure that they reproduce published results. Although EMBL-EBI established a consistent protocol for their curation, the process was manual, and no computational method or log was kept about how curators executed the models or matched published results with reproduced ones.

The BioModels repository serves as a critically important resource for the systems biology community. Using one of the many simulation tools [13] that support SBML, models downloaded from BioModels can be simulated with the expectation that results will match what was reported in the original publication. In this work, this expectation was exhaustively tested, allowing us to illuminate and correct several gaps between the expectation and reality.

To encourage SBML simulators to interpret models consistently, the SBML Test Suite [14] was developed starting in 2003 so all simulators would have a common set of unit tests that checked different aspects of the SBML specification, organized by labeled 'tags', so that when a particular simulator doesn't support a particular SBML feature (algebraic rules, fast reactions, etc.) they can flag the tests marked with that tag as not supported. Voluntarily, simulator authors may make their results public in the SBML Test Suite Database [15].

In 2008, Bergmann and Sauro [16] took a different approach towards simulator comparisons by publishing a comparison of simulator results of ten-second simulations of all 150 (at the time) curated BioModels. The differences that this work uncovered spurred discussion and activity, as simulator authors discussed the proper interpretation of certain SBML constructs, and changed their own simulators accordingly. Unfortunately, the infrastructure that allowed those comparisons only lasted through 2011, but the changes it promoted survived in the updated simulators.

**SED-ML**: SBML only describes the model; it provides no information on how the results were generated in the associated published article. The community recognized this as a significant failing because it meant that other researchers would not be able to easily reproduce the results of a published article. As a result, a new community standard, the Simulation

Experiment Description Markup Language (SED-ML) was developed [17,18]. The purpose of SED-ML is to describe how the simulation results in a given research article are generated.

SED-ML aims to capture the process of loading a model, potentially changing model values, running a simulation experiment such as a time-course simulation or parameter scan, and collecting the results as tables of data (reports) or plots. Importantly, none of these steps is connected to any particular simulation application. Thus, SED-ML provides a declarative, software-independent language for describing a simulation experiment.

BioModels has included SED-ML files for about half of the curated models; however, *caveat emptor*: they existed and were potentially useful, but had not been checked. In this work, each file's usefulness has now been improved and clearly defined. Using SED-ML, we were able to determine if a simulation was *repeatable* by running it on the original software, *reproducible* by running it on different software, and *verifiable* by comparing the results from multiple simulators. Fig 1 shows a schematic of how reproducibility is supported by these two standards.

**OMEX**: To keep the SBML, SED-ML, and other files together, they can be stored in Open Model EXchange (OMEX) files, a format standardized by the biomodeling community as a standard way of collecting model experiment files together [19]. This format is essentially a ZIP file of the collected files, plus a manifest file listing and describing each file. In 2017, the BioModels database was updated to allow each entry's collection of files to be downloaded as an OMEX file, with the manifest file noting which file was the canonical model file for that entry.

## The BioModels protocol for curating models

In order for a model to enter the curated branch of BioModels, a curator must be able to reproduce a figure from a published paper. This process starts by encoding the model into a standard format, typically the Systems Biology Markup Language (SBML). The curator then selects a figure from the publication and attempts to reproduce this figure by creating a simulation experiment using simulation software. The curator will then provide the output figure from the software engine, and a brief description of the curation work (e.g., (from BioModel 836), *"Two figures were drawn from the model based on the data provided in the paper. The values of panic intensity and protection were changed to 0.1 and 0.1 by trial and error."*). Finally, model elements are annotated, following an established procedure for annotation based on MIRIAM guidelines [20] (the Minimal Information Requested In the Annotation of biochemical Models). Once the curation is complete, all other required and auxiliary files are placed into an archive for that entry.

Although this process establishes that a set of results can be reproduced, there are some shortcomings. Curation often involves some guessing on the part of the curator, which can make the process non-repeatable if those choices are not recorded. The curation results may depend on which figure is being reproduced, the simulator settings, the time steps, initial conditions, and other considerations that must be inferred from the published article. This kind of information can be stored in SED-ML [17] files alongside the SBML model.

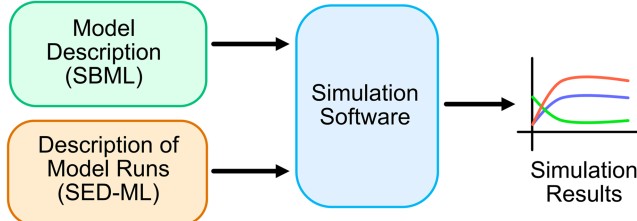

**Fig 1**. **Combining SBML and SED-ML to generate simulation results.**

## Verification of models

A number of organizations have discussed the need for improving the credibility of computational models [21–23]. The credibility of a model is strengthened if it can be verified, validated, and its uncertainty measured or quantified. These three elements are often abbreviated VVUQ, for verification, validation, and uncertainty quantification.

In this study, we focus on the verification of models described by SBML and SED-ML in the BioModels collection. It is important to note that verification does not consider whether the model is a faithful representation of the biological system under study; that would be the concern of validation. Verification is used to determine whether the numerical algorithms and model description have been implemented correctly in software. Until the development of standards such as SBML and SED-ML, automatic verification was not possible for biosimulation models, but now the conceptual description of the model (i.e. the SBML) can be processed in new computational environments by the SED-ML.

## BioSimulators

The Biosimulators site is a repository of simulation applications that support modeling standards (biosimulators.org). Each of these simulation applications is packaged as an interchangeable 'biosimulator' for running simulation experiments encoded in SED-ML. These biosimulators collectively span a broad range of simulation capabilities with enough overlap to demonstrate model reproducibility and to cross-validate predictions across multiple simulators. Each biosimulator simulation tool computes SED-ML simulation experiments for a subset of modeling formats (SBML, NeuroML, CellML), frameworks (logical, kinetic) and simulation algorithms (FBA, SSA) and are available at biosimulators.org/simulators.

The biosimulations.org website allows users to run simulation experiments specified in SED-ML, and view results and plots, and hosts a searchable collection of published simulation experiment projects, many of which were created from BioModels database models. APIs are provided for checking the correctness of SED-ML, models and metadata documents (combine.api.biosimulations.org) and for running uploaded simulation experiments and downloading the simulation results (api.biosimulations.org). A dedicated API service compares the results of multiple versioned biosimulators when exercised with the same simulation experiment (https://biosim.biosimulations.org/docs). The API was used in this work to run our simulations, and the verified results were published.

## Summary of results

In this paper, we have two main results to report. First, existing SED-ML files from curated BioModels were tested for the first time, and numerous errors were corrected. If a model had no SED-ML, we created a 'template' SED-ML file with a generic simulation experiment so that the model's behavior could be verified across different simulators. Next, with SED-ML in hand, we report on preliminary results from a verification study of all curated models across five different biosimulation platforms. This verification study allowed us to further improve the SED-ML and SBML files, as well as the simulators themselves.

To carry out this work, we took a snapshot in time from June of 2024 of the curated branch of BioModels, working with a copy of the entire curated branch of 1073 models, 1055 of which were ODE models. Of these 1055 models, we were able to create SED-ML files that produced the same results for at least two biosimulation engines in 932 cases (88%). To our knowledge, this work is the first systematic exploration of different simulators' interpretations of SED-ML constructs. Although these early results are not perfect, the use of SED-ML provides a way forward. These formal specifications can now be used to investigate the differences discovered across biosimulation engines, verifying not only the models themselves but the simulation engines as well.

This improved the reproducibility of the curated BioModels. Originally, the models were what might be termed **'historically reproduced'**: at least one curator had been able to use the model to reproduce a figure in the paper, using a different simulator than had been used in the original study. However, working instructions on how a user might reproduce the

curator's work was not able to be stored. Still, a user of the SBML file could be assured that it had been used to reproduce a figure at least once.

The addition of a 'template' SED-ML file made models additionally **'functionally reproducible'**: any user could now test different simulators to see which ones can both handle model and produce results in line with other simulators. Even though these SED-ML files do not describe how to reproduce a figure from their original publication, they do provide a common template to check simulators against each other. This work mirrors a similar study in 2008 [16] where simulations of BioModels were compared, and numerous differences between simulators were brought to light and subsequently fixed.

When SED-ML that describes the actual published experiment is available, the experiment it describes has **'end-to-end reproducibility'**: anyone with a compatible simulator can use it to follow the same procedures and obtain compatible results. Those results can then be compared by eye with the original published figures, and compared numerically across simulation platforms.

If not only figures but numeric results are available, a paper may have **'data-backed reproducibility'**: not only can the procedures be replicated in new environments, but the results can be compared with the actual values from the original study. Unfortunately, these numbers are rarely available, and researchers who wish to reproduce those results must instead rely on eyeballing the published figures, or back-calculating approximate values from pixel positions. As such, our efforts here were limited, at this time, to 'functional' and 'end-to-end' reproducibility.

These reproducibility levels are detailed in Table 1, and illustrate the increased functionality that each level provides.

## Methodology

Although there are many ways to improve the reproducibility of systems biology modeling, here we focus on the ability of researchers to replicate published simulation results and then to verify them using different biosimulation engines. As Fig 2 shows, model repositories such as BioModels that include explicit instructions as to how the results were obtained (such as SED-ML files) greatly facilitate attempts to reproduce the results obtained with those models. We have created a wrapper architecture at BioSimulators.org [24] where this information can then be interpreted and sent to different biosimulation engines. The majority of models in the curated BioModels collection are ODE models (1055 of 1073); therefore, we have selected five well-tested biosimulation engines, and built and tested wrappers for these: COPASI [25], Tellurium [26], VCell [27,28], PySCeS [29] and Amici [30]. As we describe, after many fixes and updates, many curated models (but not all) could indeed be verified across more than one of these biosimulation engines.

### SED-ML wrapper development

Most biosimulation software engines are developed independently, each with its own unique interfaces. While many offer APIs (Application Programming Interfaces) for accessing their capabilities, these APIs differ from one engine to another,

**Table 1**. Reproducibility levels.

|  | Historical | Functional | End-to-end | Data-backed |
|---|---|---|---|---|
| Model was used to reproduce a figure. | ✓ | ✓ | ✓ | ✓ |
| Model can be loaded into multiple simulators. |  | ✓ | ✓ | ✓ |
| Simulator results are numerically close to each other. |  | ✓ | ✓ | ✓ |
| Visually-close figures can be reproduced on multiple simulators. |  |  | ✓ | ✓ |
| Numerically-close figures can be reproduced on multiple simulators. |  |  |  | ✓ |

requiring customized handling of input and output data to match each engine's specific API. Our solution, described previously [24] and illustrated in Fig 2, is a "wrapper" architecture: each wrapper translates SED-ML instructions to API calls for the respective biosimulation engine, all within Docker containers to ensure platform independence and stability over time. By using this approach, SED-ML serves as a *lingua franca*, enabling the same procedure to be executed across different wrapped biosimulation engines. This allows researchers to precisely verify results using standardized inputs and outputs across multiple engines.

Each wrapper takes an OMEX file containing at least one SED-ML and one SBML file as input, translates the SED-ML instructions into simulator-specific commands, collects the output of any simulations from those simulators, and exports all outputs. Tabular data is exported as HDF5 files [31] (a format similar to CSV, but allowing multi-dimensional data and annotations), and figures are exported as PDF files. While non-semantic differences in figure output prevent the automatic comparison of figure data, the numbers inside the HDF5 files can be compared directly. In this way, we can verify whether the SED-ML files will produce the same output on multiple biosimulation engines.

## BioModels SED-ML updates

At the core of our process was a collection of Python programs, available at https://github.com/sys-bio/temp-biomodels, in concert with GitHub actions that would run the program over all 1073 BioModel curated entries. This project performed many general curation steps, but focused on updating and testing the SED-ML files. In a 'end-to-end reproducible' entry, there would be a SED-ML file present, with instructions for how to reproduce a figure from the curated paper, encoding in a reproducible way the process the original curator of the model undertook when creating and testing the model. Barring that, 'functional reproducibility' can be obtained with a SED-ML file that describes some sort of simulation of the model that could be replicated across multiple simulators.

We started with the two types of files that might be present: SED-ML files and COPASI files. Helpfully, the current version of COPASI (4.43) is backwards compatible with older COPASI files, meaning that we could create SED-ML for many entries that had been created before SED-ML even existed. When both a COPASI model file and a previously-exported SED-ML file were present for the same model, we hand-compared the existing file to one recreated from the COPASI file. In most cases, the files were nearly identical or the newly recreated files were more precise and detailed. In three cases, the existing SED-ML file had more details than the re-generated version. We hypothesized this was due to the SED-ML

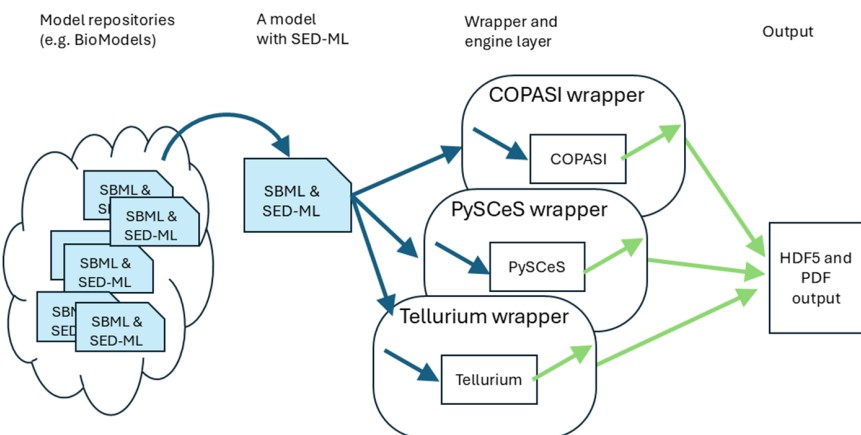

**Fig 2**. **SED-ML and wrappers that allow researchers to replicate results across multiple simulators.** See Shaikh et al. [24] for more details about the biosimulation engine wrapper architecture.

having been exported from a working COPASI curation session which was not subsequently stored as a COPASI file. The results of these hand comparisons were encoded into our Python curation program.

For the 579 entries with no SED-ML nor a COPASI file that could be used to generate SED-ML, we created a 'template' SED-ML file, encoding a simple time course experiment into it. The model is loaded, a timecourse simulation is performed for ten time units, and finally, the levels of all variable species are exported, both as a table of values per time point, and as a plot. Of course, these files do not match any figures from the original publication, but can still be used for verification purposes: if two simulators produce the same output from this template SED-ML, it shows that the model itself is robust to interpretation. It is also much easier to edit an existing SED-ML file than to create one from scratch, making future efforts to reproduce published figures simpler [32].

### BioModels simulation testing, corrections, and upgrades

A separate GitHub repository (https://github.com/biosimulations/biosimulations-runutils) was developed to run OMEX files on our wrapped simulators using the https://biosimulations.org/ API, and collect and compare the results. Unlike the temp-biomodels repository, this project was created to be generally usable: anyone can submit an OMEX file as input and compare the resulting output across simulators. We used it as part of our workflow to simulate all 1055 BioModels ODE OMEX files on five wrapped simulators, compare results, make improvements, and repeat.

When we first attempted to verify results across multiple simulators with COPASI-generated SED-ML files, almost every single run failed due to a limitation of COPASI: every SED-ML file pointed to a 'model.xml' SBML file that did not exist, since COPASI only knows about its internal model, and has no way to know what filename it might have been exported as. This is true both for the SED-ML files we generated from the original COPASI files as well as the existing legacy SED-ML files found in existing BioModels entries, making all of them incorrect apart from the five that had been created by hand. This reinforced the notion that SED-ML was being produced by curators as a perceived convenience to users but never checked, or this obvious deficiency would have been revealed.

So the next goal of our Python curation process was to add a fix to replace the fictitious 'model.xml' references with actual SBML files. When only one SBML file was present, this was trivial; when multiple SBML files were present, heuristics were developed based on file names and content similarity.

This gave us a complete, testable pipeline: SBML and SED-ML files were present that could be sent to our five wrapped simulators, and results could be compared. We could cross-verify using the native SED-ML support in Tellurium [26], separate from its wrapped version. Our process was to run a given BioModels entry across several wrapped simulators, test it in Tellurium directly, and check for error messages or empty or mismatched output. Errors would be discovered and fixed, and then we would run the entire pipeline again. This was clearly the first time anything like this had been attempted, as hundreds and hundreds of problems were discovered.

Many problems were present in the SED-ML files themselves, beyond the 'model.xml' issue: references were found to nonexistent model elements; simulations were defined and never used; duplicate elements were present; pointers to model elements were incorrectly formatted; simulation parameters were incorrectly applied; and many other issues were present. Other more prosaic issues were also discovered, such as requests for millions of output data points, which could bog down simulators for hours. As each issue caused a simulator run to fail, these issues were discovered, and fixes were added to the Python curation pipeline so fixed files could be further tested.

Occasionally, issues were found in the original SBML models. The most common issue we found were otherwise unused parameters which were mistakenly initialized to infinity or NaN. Models were also found with incorrect SBML constructs in packages such as Layout or Render (used to store visualization information in the models). Fixes for these problems were also added to the Python curation pipeline.

In some cases, we could address issues in the simulators by adding workarounds in the wrappers. For example, Tellurium had no way to adjust SBML 'local parameters' on the fly, so adjustments were made to the wrapper to translate

local parameters to global parameters, where they could then be adjusted. Similarly, PySCeS does not support SBML 'initial assignments', so we added instructions to the wrapper to translate initial assignment formulas to numerical values before asking PySCeS to simulate the model. A few of our wrapped simulators will not export constant parameters as output, so routines were added to the wrappers to add them when requested by the SED-ML.

The simulators themselves also were fixed and upgraded. The simulators worked on by our own development teams (Tellurium and VCell) were straightforward to add fixes and enhancements to, but this work also resulted in upgrades and enhancements to COPASI and PySCeS as well, as runs were found that failed on those simulators, bug reports were filed, and the development teams of those simulators produced new releases.

Other issues required more systemic solutions. As one dramatic example shows, Fig 3A and 3B show the original simulation results from COPASI and Tellurium (respectively) for Biomodel 1, a basic model with a template SED-ML file. The results for Tellurium are clearly incorrect, with several concentrations going negative. After experimentation and discussion with the COPASI developers, we discovered that while both COPASI and Tellurium have a setting called 'absolute tolerance' with a default value of $1e^{-12}$, COPASI then creates an absolute tolerance vector, using that value, but scaling it by the initial values of every variable in the simulation before sending that to the solver. When Tellurium was updated to allow the same technique with its own solver, it was able to replicate the COPASI results (Fig 3C). This information was then added to the SED-ML by creating a new 'absolute tolerance adjustment factor' setting for the simulation algorithm in the KiSAO ontology [33], which SED-ML uses to fine-tune simulation runs. The wrappers were all updated to understand and properly apply that new term, and since these results were more robust, Tellurium itself was modified to use an absolute tolerance adjustment factor by default, instead of using its old behavior.

Even this single fix has therefore improved the user experience on several fronts: model creators can encode SED-ML files more robustly with greater certainty that their results can be replicated; users of BioModel 1 (and several other models with the same phenotype) now have more varied options of simulators on which to run it; and users of Tellurium now have a more robust simulation engine for their models.

## BioModels secondary file updates

In addition to updating and creating SED-ML files, the python scripts at https://github.com/sys-bio/temp-biomodels also ensure that all the various secondary files are accurate. As one example, many BioModels included a PDF file that summarized the contents of the SBML in a human-readable form. This file is not strictly necessary for reproducibility, but as a basic quality control measure, the contents of the PDF should actually match the contents of the SBML file it is supposed

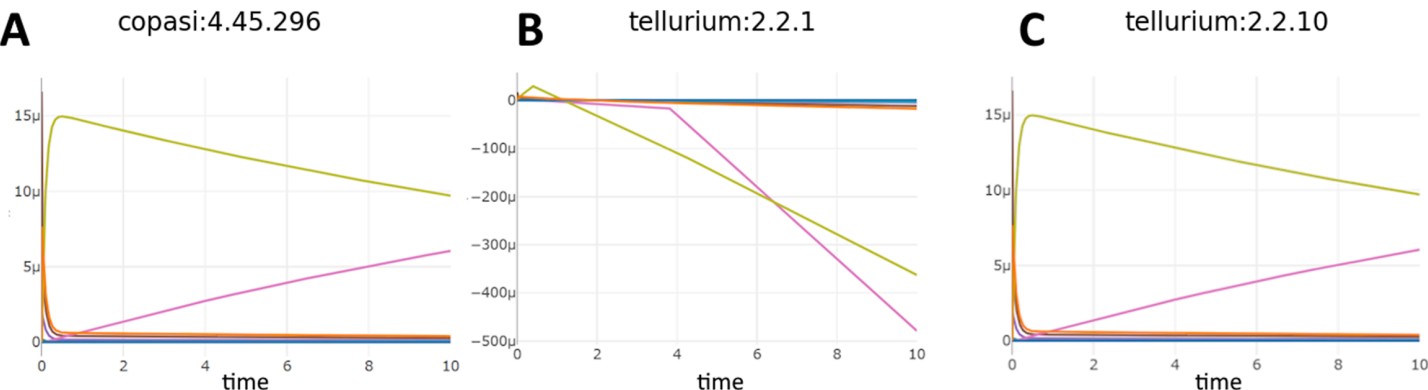

**Fig 3**. BioModels 001 simulated with Copasi (A), an older version of Tellurium (B), and the latest version of Tellurium (C).

to be summarizing. The BioModels entries contained a total of 11,890 files of 24 different types, from SED-ML files to original COPASI files to images and PDF documents. At EMBL-EBI, the BioModels curation process (described above) has generally been focused on the SBML model file, with all ancillary files treated as secondary to the SBML file. These secondary files had never before been systematically checked for accuracy, so our scripts added these checks. They also performed some basic maintenance: 40 files of size zero were discovered and removed; filenames with illegal characters were renamed; filenames of one type were renamed to a consistent scheme ('.sedml'; '.ode'); a few filename typos were fixed. We found file analyzers for all 24 file types, checked them all for legality, and fixed the ones we could. Any file auto-generated from an SBML file was re-generated with the latest version of the auto-generation program, and we generated new files for BioModels that lacked them.

Many SBML files were present in two forms: one with annotations in URN form, and another with annotations in URL form. This reflected the diversity of opinion in the SBML community when BioModels was first created as to which format was more useful. In the intervening years, that debate has been concluded, with the URL form gaining universal acceptance. Thus, the legacy URN-formatted SBML files were removed as well. Then we checked the URLs of the annotations themselves, ensuring that each pointed to an actual entry in an existing ontology. The handful of errors that we discovered here were manually corrected, consulting the original paper for references when necessary.

All of these new files (including the SED-ML files) were then pushed to the official BioModels repository at http://biomodels.org, where they all show up as the latest versions of the curated models. This means that executable SED-ML is now available for all of these models, each noting whether it is a 'template' file or one drawn from the curation process.

Moving forward, the scripts developed can be used by EMBL-EBI curators to perform more robust curation: not only can the files they collect be examined for correctness, but they can now additionally store checked SED-ML files and test them across multiple biosimulation engines on biosimulations.org. The BioModels curation protocol has therefore been updated to include these new steps (see Fig 4).

## Results and discussion

All 1073 curated entries retrieved from the BioModels database have been updated, with every file now checked for correctness or replaced. All 1055 ODE entries now have a new SED-ML file that can be used to simulate the model in some capacity: there are 579 "template" SED-ML files, and 476 SED-ML files that encode a simulation created as part of the original curation process (likely reproducing a figure from the publication).

As shown in Table 2, the great majority of these files run with the Tellurium and Copasi wrappers, and hundreds of models run successfully using our other three SBML wrappers. Tellurium has the most, simply because we focused our efforts on the wrapper for this platform since we have the most knowledge and control over this biosimulation engine. Work is ongoing to update the other wrappers to match. Fig 5 shows the distribution of successful runs and replications for both template and "full" SED-ML; i.e., SED-ML code for the 476 models from the curation process.

In Fig 5, the height of the full bar indicates the number of models that successfully ran on zero (left) through five (right) of our wrapped simulators. Within each bar, the color indicates the number of simulators whose results matched each other. Both the leftmost 'zero' and 'one' columns could only have no matches, as there were not two successful simulations to compare against each other. For the most part, the 'template' SED-ML entries in Fig 5A ran on more simulators than the 'full' SED-ML entries in Fig 5B, as the latter use a wider variety of SED-ML features, not all of which are supported by all the wrappers or simulators.

For all models with successful runs on at least two wrapped simulators, we defined a successful replication as having results where every element matched to a relative tolerance of 0.0001 and absolute tolerance scaled by the range of output values for that variable, using the numpy 'allclose' algorithm [34]. As can be seen, of the 'template' SED-ML files, all but one of them successfully ran on multiple simulators, and of the 'full' SED-ML file entries, one did not run on any of the five simulator wrappers, and 22 only ran on one. These 24 models were unable to be verified because two sets of

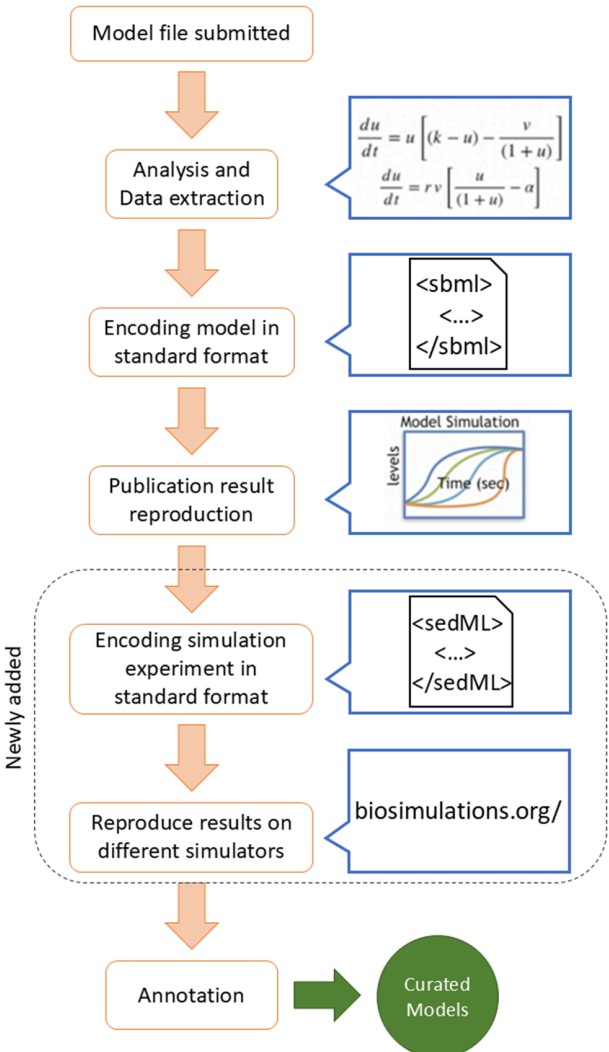

**Fig 4.** **The revised BioModels curation protocol.** Adapted from Malik-Sheriff, et al., 2020 [7] with permission.

**Table 2.** **Successful runs for each simulator of the 1055 currently curated ODE-based BioModels.**

| Simulator | Successful runs |
|---|---|
| Tellurium wrapper | 1036 |
| Copasi (basico) wrapper | 1032 |
| VCell wrapper | 893 |
| Amici wrapper | 758 |
| PySCeS wrapper | 711 |
| At least two of the above | 1031 |

results did not exist to be compared. Of the remaining entries with the relatively straightforward template SED-ML, replication was possible in 578 cases, with 'functional reproducibility' verified in 546/578 cases (94%). For the potentially more complicated entries with full SED-ML, 453 cases could be replicated, with 'end-to-end reproducibility' verified in 386/453

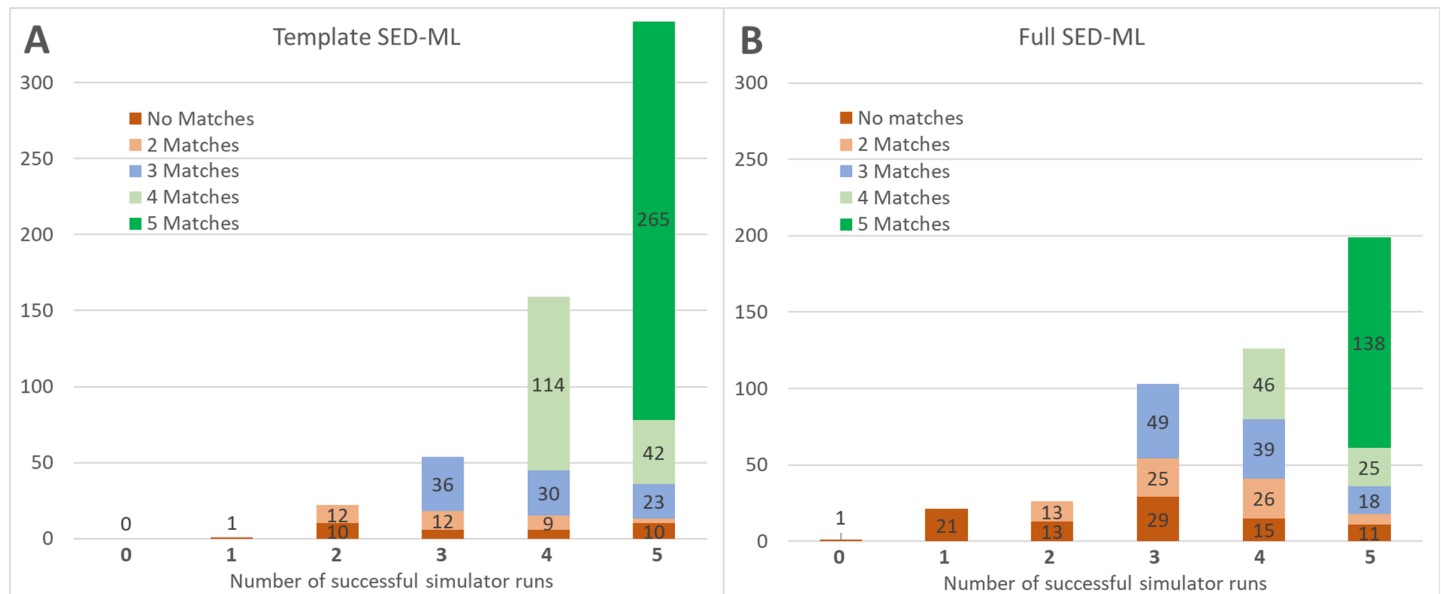

**Fig 5**. Template (A) and Full SED-ML (B) replication across five wrapped simulators.

cases (85%). Overall, of the 1055 updated BioModels, we were able to verify functional or end-to-end reproducibility for 932 models (88%).

Initially, our results for replicability across engines were much weaker. An earlier draft of this manuscript showed much lower success rates for simulation engines other than Tellurium. However, once these early results were circulated among the community, a broad and significant effort was invested by many institutions and research groups. Bugs were fixed and improvements were made to the SED-ML files themselves (such as fixing references or ranges), the biosimulation wrappers (such as fixing SED-ML interpretation, and improving edge case handling), the infrastructure that runs the wrappers (such as improving latency and robustness), and the simulation engines themselves (such as improving SBML interpretation and handling of simulation edge cases). Thus, an important contribution of our work has been to stimulate this type of community-based work to improve the consistency and accessibility across a range of simulation engines. This work is not finished, and is expected to continue into the future, spurring further improvements.

All of these improvements are freely available: the SED-ML files are now available at part of the BioModels entries at http://biomodels.org, the wrappers are available on github at https://github.com/orgs/biosimulators/repositories, the Docker images are available at https://biosimulators.org/simulators, the wrappers can be run online at https://biosimulations.org/runs/new, or by using our Python package https://github.com/biosimulations/biosimulations-runutils, results from verified BioModels entries (and others) can be viewed and downloaded from https://biosimulations.org/projects, and the improved simulators can all be accessed from their authors' web sites. Table 3 shows a handful of fixes and enhancements to some of the software packages we've been working with, though it is by no means an exhaustive list. Full lists of changes can be found in the commit histories of the individual projects.

Additional improvements in consistency will now be easier to carry out. Simulators that fail to produce results from particular BioModels entries now have a test harness where improvements to the simulator or simulator wrapper can be checked as they are expanded to handle a wider variety of inputs. Cases where a simulator's results for a given BioModel entry did not match replicated results from two or more other simulators can now be pulled out and examined: at least 30 such models exist for every single simulator in our study.

**Table 3**. Example software fixes and enhancements.

| Software | Fixes and Enhancements |
|---|---|
| Tellurium wrapper | • Convert local parameters to global parameters so they can be changed by Tellurium's native interface.<br>• Fix bug with referenced values in nested repeated tasks.<br>• Handle SED-ML output start times different from initial times. |
| PySCeS wrapper | • Correctly output variables controlled by rules.<br>• Convert initial assignments to values before handing model to PySCeS (which doesn't support initial assignments). |
| VCell | • Allow import of models with reserved words as ids. |
| Copasi | • Fix handling of reactions with parentheses in their names. |

Our work also allows future BioModels curators to avoid these problems from the beginning. Using the scripts we created for this project, curators will be able to generate and test SED-ML instructions more easily, and verify results across simulators (see Fig 4).

Recall that this study stops short of ensuring that the simulator runs actually reproduce a figure from the original publication; for many models, all we have is brief "curator comments", so this process cannot be easily automated. We expect that most of the 476 entries (see Fig 5B) that contained SED-ML or COPASI files will indeed match a figure from the paper, as those files were originally created during curation. However, this is not guaranteed, so every model will need to be checked by hand. The entries with template SED-ML will not match published results, and the SED-ML would need to be modified to match the original computational experiment.

It is now also increasingly possible to usefully critique the models themselves: how reproducible are they? How sensitive are the published results to the parameter values? Can they be extended? With a system in place that makes re-simulation and verification straightforward, these questions become much easier to answer.

## Conclusions and future directions

This project demonstrated that even manually curated models can be challenging to reproduce without appropriate specifications for how to carry out the experiment. We demonstrate the utility of the SED-ML language for capturing these specifications by comprehensively creating or improving SED-ML files for 1055 ODE-based models in the BioModels database, all but one of which could be successfully run in at least one of our five wrapped simulation engines. For 88% of the models (932/1055), we were able to show that two separate simulators produced the same results within reasonable tolerances. Work still needs to be done to fully replicate the original curators' work in reproducing the published results, as there is no formal record of how these models were originally curated. We have begun some of this work, and early results are encouraging [32].

Determining the scope of what remains to be done is hard to discern, but the general shape can at least be roughly determined. Many issues are already known: models that failed to run with a particular wrapper (845 runs altogether, compared to the 4430 successes) all have error messages that can be examined. Determining the problem with runs that failed to match any other simulator is more complicated, as there is no obvious error message to start with. Here, there are 1624 pairwise comparisons that failed, compared to 6016 comparisons that succeeded. (Remember that the more simulators that successfully produce output, the more possible pairwise comparisons there are: when all five produced output, 10 comparisons are possible.)

In many cases, the model in question contains a feature of SBML that is not supported by the simulator. For example, Amici does not support delayed events (used in 56 BioModels), neither Amici, PySCeS, nor VCell support assigned stoichiometries (used in two BioModels), and VCell does not support non-integer stoichiometries (used in 33 BioModels). Considering solely the SBML features tested in the SBML Test Suite, this explains 174 of the 845 runs that failed (21%), plus an additional 92 of the 1624 failed pairwise comparisons (5.7%), for cases where a simulator is known to not handle

certain inputs, but still produces output. As a specific example, there are 7 models in BioModels that use delay differential equations. None of our five tested simulators support this construct, but Copasi and PySCeS produce output anyway. These outputs do not match each other, and they certainly do not match the modeler's intent.

Other failures are due to the SED-ML using features not supported by the wrapper. The VCell wrapper, for example, does not support repeated tasks or steady state analyses (91 of the 'Full' SED-ML models), and other wrappers similarly supported only a subset of those analyses. This explains why the 'template' SED-ML had better success rates than the 'full' SED-ML: the latter was more likely to use uncommon features of SED-ML that our wrappers did not universally support.

For some comparisons, our criteria may have simply been too strict: the magnitude of the variables involved and the scope of the model itself might have warranted a less stringent tolerances. Of the 1624 failed pairwise comparisons, 378 of them (23%) result in a match score between 1 and 10, where '1' is the maximum cutoff for whether two data sets match or not. These models are good candidates for further examination, since slightly different tolerances could have resulted in a match. Each such comparison would need to be examined by hand to determine whether this was the case or not, but false negatives that can be examined by hand are much easier to find than false positives that seem to match but actually have hidden significant differences.

While a complete analysis of the remaining failures and mismatches will take time, we did a 'spot check' of the 21 models which every simulator successfully produced results for, but no two models matched. For thirteen of these, all five visually looked 'the same', even though their match scores were too high. For the remaining eight, at least two of them visually matched, but there were obvious mismatches with the rest. Over the whole set, half of them had oscillations or other rapid changes in values, which could in theory explain some of the results that looked the same but scored poorly: very slight phase differences between results could end up with significantly different numerical values. For oscillating output, therefore, a different method of comparison may be necessary. Fig 6 shows two sets of results where no pair of results passed our 'match' criteria. For the first (A) from BioModel 19 [35], Amici and VCell's results seem to be close to each other, but different from the others, and Copasi and Tellurium's results are similarly close to each other, but different from the rest. For the second (B) from BioModel 117 [36], all five are close, but contain the rapid oscillations that may have prevented them from passing our 'match' test. Determining which results from BioModel 19 are intended, and which represents the most accurate interpretation of the SBML model they all purport to simulate will take further research. Similarly, discerning whether the small differences from the BioModel 117 results (expanded in Fig 6C) are significant or trivial will necessitate revisiting the original paper.

But whether or not the failures can be categorized, it is simply often the case that hidden problems are only revealed when something is explicitly tested. A major contribution of this work is simply the testing framework it provides, for both the simulators we tested and the simulators we did not.

This approach can also be extended to modeling paradigms besides ODEs: SBML itself has the ability to store and standardize different models like flux balance constraint (FBC), spatial, and qualitative models. In addition, SED-ML is designed to be flexible enough to store computational experiments for those model types, and multiple simulators exist that can perform those experiments and whose results can then be compared. Thus, if instructions for experiments can be encoded in a standard way for standardized models (perhaps using SED-ML and SBML), they could be packaged in OMEX files, sent to different simulators, and their outputs compared. The most difficult part of this process will be bridging the gap between the instructions and the simulators: native support for SED-ML could be added to the simulators themselves, or new wrappers could be developed that translated those instructions, as in this study. Efforts along these lines have already started, with the introduction of the FROG analysis for FBC models [37], in which the model's 'Flux variability, Reaction deletion, Objective function, and Gene deletion' are tested. As of yet, no harness exists to automatically produce the desired outputs, but the instructions, at least, are standardized. Similar comparisons with FBC models across simulators [38] have spurred debates [39,40] regarding defining the standards and solver capabilities that together determine whether results from a flux balance model analysis can be reproduced.

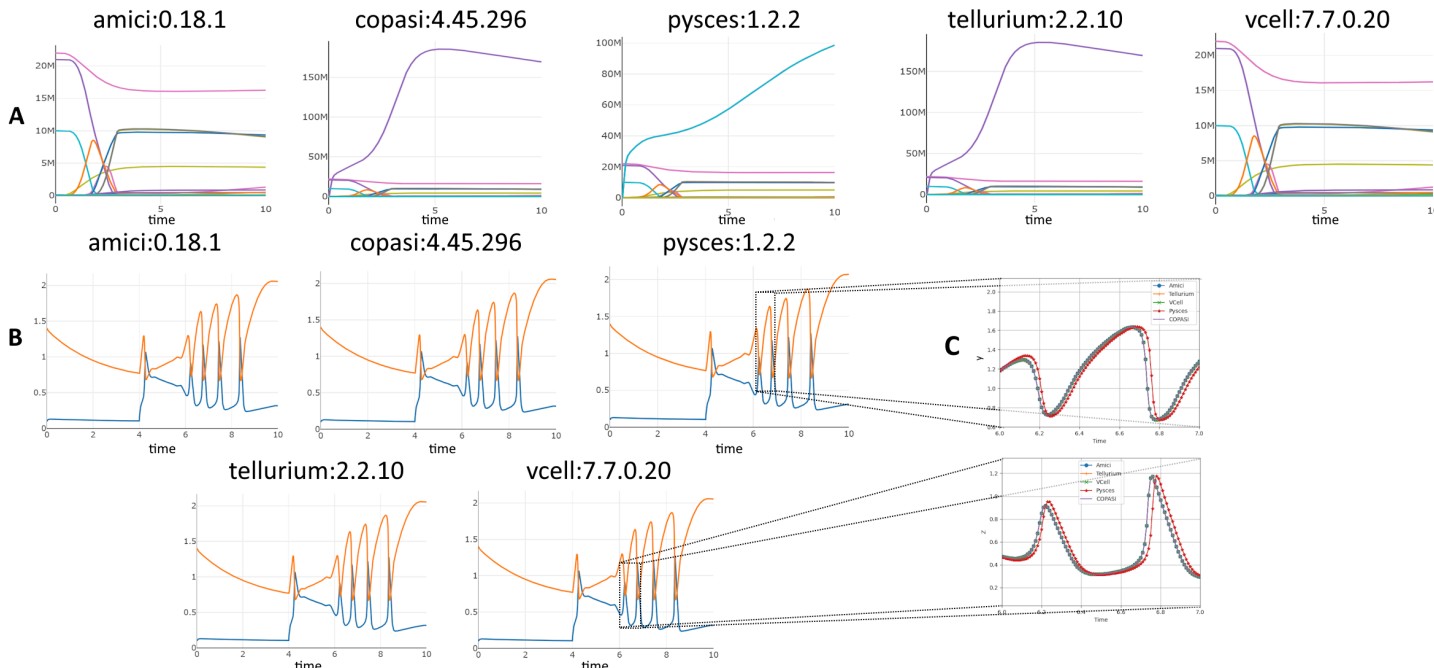

**Fig 6**. **Simulation results from BioModel 19 (A) [35] and from BioModel 117 (B) [36], from all five wrapped simulators. The two graphs in C show a zoomed-in subset of the data from BioModel 117: all five simulator results on a single graph for y (top) and z (bottom).** No pair of simulator results passed our 'match' threshold, for reasons that are obvious to the eye for Biomodel 19, but for reasons that can only be seen at larger scales for BioModel 117.

Moving forward, this project demonstrates the need for improved SED-ML generation tools. Model-building environments should be able to generate SED-ML for any given experiment or execution of a model. The COPASI tool comes close to this, but it only creates SED-ML for a single model/experiment/plot combination, which isn't sufficient for experiments involving multiple models, nor plots involving multiple experiments. In addition, it cannot export model changes (i.e. 'in this figure, the value of n was 2, and the value of S1 was 2.42'). As an alternative approach, the Tellurium Python environment includes the phraSED-ML library, which makes it easy to create and execute a SED-ML file, but 'creating a SED-ML file' has to be a goal of the user, who otherwise will just use Python directly to carry out their experiments. For other modeling paradigms, 'creating the SED-ML file' will be a similarly difficult challenge.

In general, auto-generating SED-ML will be challenging. In a GUI environment like COPASI, model changes are executed and not stored, so an experimental protocol that includes 'change this value' will not encode that change into the SED-ML, but will instead export a different initial model with those new values. In a scripting environment like Tellurium, common scripting behaviors like loops will commonly be encoded into the scripting environment itself, making it difficult to then export to SED-ML. Any tool that produces SED-ML but requires the user to encode these procedures separately from the user's usual workflow must therefore offer significant benefits in exchange.

Finally, an important contribution of our work is to demonstrate the value of the wrapper architecture for biosimulation engines, and how that can support model verification. As we have shown, although many models do produce the same results across multiple engines, some models do not execute on some engines, and some models produce different results across engines. All of our scripts and wrappers are available via GitHub (https://github.com/biosimulators/), and we hope that model builders will be interested in using these to test their experiments across multiple biosimulators. Should they do so, those OMEX and SED-ML files would essentially reduce the time spent curating their models to zero In the

future, we plan to build a REST API for verifying any model coded in SBML (and/or OMEX archives with SBML) across multiple biosimulation engines.

Although the BioModels database has carried out ground-breaking work developing a curated model repository, until now, the database has not included enough information to fully replicate published results. Here, we have made significant inroads towards solving this problem by providing SED-ML files for all models, as well as SED-ML files involved in curation for almost half of the collection. Moving forward, additional work on improved wrappers is needed, and model development environments should be designed to facilitate the auto-generation of SED-ML. Without strong tool support and systematic checking of results, it is too easy for errors or omissions to accumulate, which ultimately makes published results non-reproducible. These scripts and services are available to everyone, not just curators, so model developers themselves can use them to ensure the reproducibility of their own models. We strongly encourage authors to submit their models, accompanied by correct SED-ML files, to BioModels or comparable resources as part of the publication process. With the use of SED-ML, we have demonstrated the ability to *verify* published results across multiple biosimulation engines. This increase in reproducibility has in turn increased the value of the repository to modelers going forward. New models will be directly comparable to existing ones using the same protocols, and the existing models can be expanded and used in new contexts with greater assurance of their utility.

After verification through reproduction, the next goal of modeling is to pass validation: the successful comparison of computational data to biological data. In order for this to be added to the curation process, the original biological data would have to be obtained (hopefully from the supplementary materials from the original publication), stored in a standard format for inclusion in the OMEX file, and instructions added (perhaps in SED-ML) to compare those data with the computational experiment results. This has been beyond the scope of the curation process, but could be particularly valuable in the future, as it would allow fully-reproducible VVUQ (Verification, Validation, and Uncertainty Quantification) analysis, which is currently not possible.

## Acknowledgments

This work would not have been possible without the years of work from over 200 BioModels curators and modelers who submitted their work to the BioModels repository.

## Author contributions

**Conceptualization:** Lucian P. Smith, Jonathan Karr, Ion I. Moraru.

**Data curation:** Lucian P. Smith, Rahuman S. Malik-Sheriff, Tung V. N. Nguyen, Jonathan Karr.

**Funding acquisition:** Ion I. Moraru, John H. Gennari, Herbert M. Sauro.

**Investigation:** Lucian P. Smith, Jonathan Karr.

**Methodology:** Lucian P. Smith, Rahuman S. Malik-Sheriff, Tung V. N. Nguyen, Jonathan Karr, James C. Schaff, Joseph L. Hellerstein, David P. Nickerson.

**Project administration:** Lucian P. Smith.

**Software:** Lucian P. Smith, Jonathan Karr, Bilal Shaikh, Logan Drescher, James C. Schaff, Alexander A. Patrie.

**Validation:** Lucian P. Smith.

**Writing – original draft:** Lucian P. Smith, John H. Gennari, Herbert M. Sauro.

**Writing – review & editing:** Lucian P. Smith, Rahuman S. Malik-Sheriff, Henning Hermjakob, Jonathan Karr, Ion I. Moraru, Eran Agmon, Michael L. Blinov, Joseph L. Hellerstein, Elebeoba E. May, David P. Nickerson, John H. Gennari, Herbert M. Sauro.

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
