## [Decision Letter · Decision Letter 0]

3 Aug 2025

PCOMPBIOL-D-25-01206

Verification and reproducible curation of the BioModels repository

PLOS Computational Biology

Dear Dr. Smith,

Thank you for submitting your manuscript to PLOS Computational Biology. After careful consideration, we feel that it has merit but does not fully meet PLOS Computational Biology's publication criteria as it currently stands. Therefore, we invite you to submit a revised version of the manuscript that addresses the points raised during the review process.

Please submit your revised manuscript within 60 days Oct 03 2025 11:59PM. If you will need more time than this to complete your revisions, please reply to this message or contact the journal office at ploscompbiol@plos.org. Please include the following items when submitting your revised manuscript:

We look forward to receiving your revised manuscript.

Kind regards,

Marc Birtwistle

Section Editor

PLOS Computational Biology

**Journal Requirements:**

At this stage, the following Authors/Authors require contributions: Lucian P Smith, Rahuman S. Malik-Sheriff, Tung V. N. Nguyen, Henning Hermjakob, Jonathan Karr, Bilal Shaikh, Logan Drescher, Ion I. Moraru, James C. Schaff, Eran Agmon, Alexander A. Patrie, Michael L. Blinov, Joseph L. Hellerstein, Elebeoba E. May, David P. Nickerson, John H. Gennari, and Herbert M. Sauro. Please ensure that the full contributions of each author are acknowledged in the "Add/Edit/Remove Authors" section of our submission form.

4) We note that your Data Availability Statement is currently as follows: "All relevant data are referenced within the manuscript.". Please confirm at this time whether or not your submission contains all raw data required to replicate the results of your study. Authors must share the “minimal data set” for their submission. PLOS defines the minimal data set to consist of the data required to replicate all study findings reported in the article, as well as related metadata and methods (https://journals.plos.org/plosone/s/data-availability#loc-minimal-data-set-definition).

6) Please send a completed 'Competing Interests' statement, including any COIs declared by your co-authors. If you have no competing interests to declare, please state "The authors have declared that no competing interests exist". Otherwise please declare all competing interests beginning with the statement "I have read the journal's policy and the authors of this manuscript have the following competing interests"

**Reviewers' comments:**

Reviewer's Responses to Questions

**Comments to the Authors:**

Reviewer #1: This manuscript from Smith and others claims to have generated a revised Biomodels database collection that contains verified and reproducible models, whereas before the database contained many models that were not so. The primary definition they use is that each model in SBML form should be able to be paired with a SED-ML document for specifying simulations, and then simulated by at least two different tools to obtain the same time course information. A main issue is that this definition of a reproducible model seems to fall short of what many would consider to be, is that the model in the database reproduces the original claims of the model and/or simulation data given in the original publication. The fact that an SBML model and a SED-ML paired document might give different results using different simulators seems much more likely due to the simulator having issues, and /or the wrappers that are used to convert the information to the format needed for the simulator. In fact, in one of the few author’s examples (Fig. 3), this is the case. Thus, the manuscript as written does not seem to support the very broad claims being made. That being said, the set of wrappers written to interface a common SBML+SED-ML file set to a variety of simulation tools could be a useful contribution to the modeling community, and that in this reviewer’s opinion is one of the novelties of this work. However the fidelity of these wrappers is uncertain, given the extremely surprising lack of agreement between many simulators, which seem to all be using standard ODE solvers. Some more focus on why so much disagreement is seen would be very important and is quite disturbing. Specific comments on the paper are further given below:

• If separate simulation engines do not produce the same result, it may not be the issue of the model but could be an issue with the software, or a numerical methods issue. Therefore, this criteria may be too aggressive. In fact, the result highlighted in Figure 3 suggests this is the case. Why would someone want to simulate a model with multiple ODE solvers to prove reproducibility?

• Reference to some of the debate that went on with regards to metabolic modeling and discussion in the introduction would be quite helpful

o 10.15252/msb.20156548

o 10.1038/ncomms5893

o 10.15252/msb.20156157

• The introduction is generic and more detail on what problems reproducibility of models brings, and how people have already tried to solve it, would be appropriate.

• The introduction conflates to some extent open source (accessible) versus reproducibility; this distinction deserves some nuance.

• A figure which shows the EMBL protocol for model curation, versus the improvements in this work, would be helpful to the reader.

• In the introduction, the section on BioSimulators is informative but how it is relevant to the described work and how it is used could be better described.

• The possibility that the developed wrapper codes to enable SED-ML to be interpreted by different solvers play a role in observed errors was not adequately considered. That being said the development of these wrappers seems to be a large part of the novelty of this work but the way the manuscript is presented does not capture this very well.

• Line 185, the paragraph about SED-ML updates, was unclear. If the update happened back in 2021, why is it relevant to this 2025 paper?

• Line 210, what problems, and how would they be fixed? More broadly in this section, how do secondary files fit into the main point of the manuscript of computational reproducibility? Related, Line 214, it was unclear what translation programs did or why they were important.

• The usefulness of the template SED-ML files for 579 entries with respect to reproducibility of a model for the reported results was unclear, but this seems to be the major point of the paper. More impactful would be SED-ML files that proved reproducibility of results.

• It was unclear how many of the 1055 models are reproducible in terms of the claims and/or results of the publications.

• Why was no attention paid to the non-ODE models?

• Figure 4 is hard to interpret--why are there six different bars but also five different colors in each bar? One wonders as well what is the significance of the template versus full sed-ml comparison. Lastly, there seems to be an unexpectedly large amount of mismatch if the figure is being interpreted correctly, which is very surprising and deserves more analysis. This figure would benefit from changing and/or a much more detailed legend.

• Line 381 paragraph, this work that went into changing and updating simulators seems perhaps significant but is not deeply described or enumerated in the paper. More of this would be relevant in this reviewer’s opinion.

• The fact that this study does not compare simulation results to original results undermines the claim of general reproducibility, and some of the argued significance of the work. This seems to be driven by a strict reliance on SED-ML as a component of reproducibility, which could be overly demanding. How the Biomodels “increase in reproducibility has in turn increased the value of the repository to modelers going forward” (line 478) was not completely convincing.

• Table 1 has perhaps one of the most surprising results that a large number of simulators are different. Given these are using a standard ODE solver, this is alarming, but the reasons why are unclear.

Reviewer #2: This work represents a valuable addition to both BioModels and the systems biology community.

I found the terminology usage around "validation" somewhat inconsistent. In the BioSimulators paragraph, the term appears to refer to validating the SED-ML/SBML format, which differs from your earlier definition where validation involves determining the biological accuracy of a model. While I appreciate your explicit definitions of terms in the Verification of Models paragraph, I believe you should also have provided a definition of 'translation' as its meaning was unclear in the context of 'translation programs'.

The verification results are particularly interesting. It would have been valuable to include a similar analysis using the SBML test-suite, which is widely used by simulation engine developers to verify their implementations. Your results highlight persistent issues with different applications correctly simulating SBML. The fact that very few models verified across all five simulation packages indicates significant work remains to be done—as you rightly acknowledge. I anticipate that extending your template SED-ML files will likely reveal additional issues.

Minor issues:

- The curation of BioModels has historically involved broader collaboration than just the EBI. Initially, researchers from the University of Hertfordshire and Caltech were also involved.

- Regarding standards chronology, SED-ML was not strictly the second standard to emerge, as SBGN predates it.

- The sentence on line 83, page 4 ("However, many such simulators...") appears redundant and could be removed.

- On line 373, page 11, there is a repetition of the word 'only' that should be corrected.

Reviewer #3: This manuscript represents a valuable contribution to model reproducibility and reuse. The authors looked at *all* curated ODE models in the BioModels database (1,055 models) and generated a SED-ML description of a simulation protocol for each. They then ran simulations using five different ODE solvers/simulation engines, namely COPASI, Tellurium, VCell, PySCeS, and Amici. Sometimes, this failed. And when there was run success there was sometimes a consistency problem. The authors tried to fix these problems. After this effort, they were able to get a 98% run success rate (1,031/1,055) and a “verification success” flag for 928 or 932 models (88%). Verification success was declared when two or more simulators (given the same simulation instructions) generated indistinguishable results. This report demonstrates some of the challenges of reproducing modeling/simulation results. I think it has major strengths, which are listed below. I think this report will be of interest to many in the biological modeling community. The weaknesses/limitations of the study that I note below do not require major manuscript revisions. Mostly, these points should just be appropriately recognized in the Discussion section.

Strengths

1) I’m not aware of another effort that considered all the curated models in the database. I think other related studies have focused on a selected subset. The comprehensive scope is perhaps unique.

2) I think the authors’ efforts make an impact on reproducibility. They were able to get 98% of the models running (on at least two simulators) and a verification success rate of 88%. Now other researchers will benefit from this work.

3) The authors are sharing resources freely, such as APIs.

Weaknesses

4) There was verification failure for 12% of the models, even after curation. The causes are discussed (on ~2 pages) but perhaps the analysis could go further? Could the authors say more about why they think so many models have this problem?

5) In this study, there was a focus on ODE models. There are other types of models. Could the authors say more about how their efforts might be extended to encompass other model types? What challenges would have to be overcome to consider logic/Boolean models, PDEs, agent-based models, etc.?

6) The “template” SED-ML files provided by the authors are not for reproducing published figures. This is a disappointment, but it’s understandable. Could the authors say more about why they did not go further? What are the challenges to reproducing results in figures? How could these be overcome? Can model developers do more to promote reproducibility?

7) The manuscript does not spell out the audit trail for fixes. Could the authors better explain the extent of logging and how other researchers can access available documentation of fixes?

8) There was no attempt to verify whether models capture underlying biology. We really want that type of effort, right? How can we get there? Could the authors say more about this point?

9) Success results are reported inconsistently, 932/1055 in some places, and 928/1055 elsewhere. This inconsistency should be fixed.

10) Verification success rate was 94% for 578 “template” protocols but only 85% for 453 “full” protocols. Are more complex SED-ML files a problem? The simulators evaluated don’t fully support SED-ML features?

**Have the authors made all data and (if applicable) computational code underlying the findings in their manuscript fully available?**

Reviewer #1: Yes

Reviewer #2: Yes

Reviewer #3: Yes

PLOS authors have the option to publish the peer review history of their article (what does this mean?). If published, this will include your full peer review and any attached files.

Reviewer #1: No

Reviewer #2: No

Reviewer #3: No

**Figure resubmission:**
---

## [Decision Letter · Decision Letter 1]

14 Nov 2025

Dear Dr. Smith,

We are pleased to inform you that your manuscript 'Verification and reproducible curation of the BioModels repository' has been provisionally accepted for publication in PLOS Computational Biology.

Best regards,

Marc Birtwistle

Section Editor

PLOS Computational Biology

Reviewer's Responses to Questions

**Comments to the Authors:**

Reviewer #1: Overall the authors have done an adequate job addressing the previous concerns of this Reviewer. Only some minor considerations remain that could be addressed without another round of revision.

1. Verification has a few different definitions throughout the manuscript and this could lead to confusion. For example, pg 3 says “verification to mean that separate simulation engines produce the same results when running the same computational experiment.” Pg 5 says “We use the definition of verification from the Los Alamos National Laboratory white paper [23], which defines verification as ‘. . . the process of determining that a model implementation accurately represents the developer’s conceptual description of the model and the solution to the model.”, and then this is further clarified later.

2. On pg. 7 they state “" As Figure 2 shows, as a first step, model repositories such as BioModels should include SED-ML to describe the steps needed to replicate the results." This seems overstated—a suggestion that inclusion of files such as SED-ML in BioModels and similar databases would help with reproducibility seems more appropriate.

Reviewer #2: Thank you for addressing my comments and those of my fellow reviewers. The manuscript certainly flows better.

Reviewer #3: The authors have substantively and adequately addressed my concerns.

**Have the authors made all data and (if applicable) computational code underlying the findings in their manuscript fully available?**

Reviewer #1: Yes

Reviewer #2: Yes

Reviewer #3: Yes

PLOS authors have the option to publish the peer review history of their article (what does this mean?). If published, this will include your full peer review and any attached files.

Reviewer #1: No

Reviewer #2: No

Reviewer #3: No

---

## [Editor Report · Acceptance letter]

PCOMPBIOL-D-25-01206R1

Verification and reproducible curation of the BioModels repository

Dear Dr Smith,

I am pleased to inform you that your manuscript has been formally accepted for publication in PLOS Computational Biology. Your manuscript is now with our production department and you will be notified of the publication date in due course.

With kind regards,

Anita Estes
